# Enhance Fruit Ripening Uniformity and Accelerate the Rutab Stage by Using ATP in ‘Zaghloul’ Dates during the Shelf Life

**DOI:** 10.3390/foods10112641

**Published:** 2021-10-31

**Authors:** A. A. Lo’ay, Rania E. Elgammal, Haifa A. S. Alhaithloul, Suliman M. Alghanem, Mohammad Fikry, Mohamed A. Abdein, Dalia M. Hikal

**Affiliations:** 1Pomology Department, Faculty of Agriculture, Mansoura University, Mansoura 35516, Egypt; rano90@yahoo.com; 2Biology Department, Collage of Science, Jouf University, Sakaka 72388, Saudi Arabia; haifasakit@ju.edu.sa; 3Biology Department, Faculty of Science, Tabuk University, Tabuk 47731, Saudi Arabia; s-alghanem@ut.edu.sa; 4Department of Agricultural and Biosystems Engineering, Faculty of Agriculture, Benha University, Toukh 13736, Egypt; moh.eltahlawy@fagr.bu.edu.eg; 5Biology Department, Faculty of Art and Science, Northern Border University, Arar 97911, Saudi Arabia; abdeingene@yahoo.com; 6Nutrition and Food Science, Home Economics Department, Facullty of Specific Education, Mansura University, Mansoura 35516, Egypt; dr.daliahikal@mans.edu.eg

**Keywords:** date, shelf-life, Rutab process, ATP, Zaghloul

## Abstract

The Rutab date involves a physiological process by which the fruit turns completely ripe. The objective of this study was to research the effect of ATP-treated fruit to improve their biologically active compounds of the Rutab process of the ‘Zagloul’ date during shelf-life. Fruits at full color (red) were dipped in 0, 1, 1.5 mmol L^−1^ ATP solution for 10 min, and then stored at room temperature (27 ± 1 °C) with a relative humidity of (67 ± 4 RH%) for 12 d. We found that ATP treatment, especially at 1.5 mM, enhances the Rutab stage of date fruit, and certain biologically active compounds such as total phenols and flavonoids, in all ATP treatments compared to untreated fruits. ATP enhanced the loss of tannin compounds in fruit but had no impact on the change in fruit moisture percentage of fruit during storage. The treatments did affect the changes in total sugar content and activated the sucrose enzymes, i.e., acid invertase (AI), neutral invertase (NI), sucrose synthase-cleavage (SS-c), and sucrose synthase-synthesis (SS-s) during storage. Interestingly, immersion in 1.5 mM ATP forced the date fruit to reach the Rutab stage during storage. These results indicated that the dose of ATP (1.5 mM) is a new potential tool that pushes the fruits to regular ripening after harvest, thus reducing the losses in the fruits during the production process. A linear model could be satisfactorily used for predicting the properties of the treated date with ATP 1.5 mM at different shelf-life durations.

## 1. Introduction

The date palm is the most productive and important subsistence crop in the hottest and driest desert regions [1]. The date fruit (*Phoenix dactylifera* L. CV ‘Zaghloul’) is a favorite Arab food from the Middle East [2,3]. Originally, it was cultivated in Egypt around 13,000 BC, and was later submerged in pharaoh temples [4]. Egypt now has a harvested area of 48,031 hectares, which equates to 1,603,762 tons harvested annually [5]. Dates are typically harvested and marketed at three stages of maturity: mature firm (completely red), fully ripe (Rutab), and dry (Tamer). Harvest at one stage relative to another is determined by the characteristics of the cultivars, in particular soluble tannin levels, as well as environmental conditions and market demand [3,6]. The consumption of fruits and vegetables is widely considered important and healthy. A high concentration of dietary compounds known as polyphenols is responsible for the antioxidant and antimutagenic properties of date palm in vitro [7]. Among arid and hot regions, the date palm is the most successful and most important subsistence crop [8].

Due to an uneven ripening of the fruit, multiple harvests (10–15 times) are required during the harvest season (from September to November). However, only 30–40 percent (Biser stage, red) of the total fruit will ripen on the trees, resulting in economic loss. To accelerate the ripening of the fruits, a post-harvest treatment is necessary. Fruit maturation tends to slow as temperatures decrease in September, October, and November. Such maturation failures explain the dramatic temperature drop that has occurred over this period. According to customer demand, the fruits are delivered at Khalal, Rutab, and Tamer maturity in the Egyptian market [9]. The fruit begins losing water at harvest to increase the accumulation of sugar [10]. Also, it loses in the Rutab stage more quickly, resulting in softening [11]. Numerous studies have been conducted on date fruit to maintain fruit quality or minimize fruit weight loss by using gibberellin and Ethel to improve fruit size [12]. The implementation of ethephon to control fruit thinning [13]. An alternative is to increase yields and improve quality by using 5-aminolevulinic acid [14].

Adenosine-5-triphosphate (ATP), the most direct source of energy within the cell, plays an important role [15]. It mediates various chemical reactions like a signal molecule. Extracellular ATP, for instance, increases cytosolic Ca^2+^ and leads to the production of second-signal molecules, including NO, Ca^2+^. For example, it reacts with reactive oxygen species (ROS) to regulate the defense response to biological stress [16]. Fruit firmness is provided by the ions NO and Ca^2+^ [17]. Treatment with exogenous ATP can accelerate ATP formation in plants and increase the content of fatty acids that are produced [18]. Then, it maintains the integrity of the cell membrane by delaying fruit spoilage [19]. This suppresses the respiration rate [20], as well as regulating ROS, and improves fruit quality [21].

The exogenous application of ATP increases intracellular ATP content and energy charge (EC) levels. This inhibits respiratory intensity and the browning of longan fruit and maintains a higher quality of fruit [20,22]. During storage, exogenous ATP alleviated the quality breakdown and browning of mung bean sprouts by regulating ROS metabolism and pectin degradation [21].

When date fruits reach the ripening stage, many of them fall off the cluster. As the date fruit growth curve is sigmoidal, by the end of the cell elongation stage known as the Kimri stage, the fruits may vary in the development stage. As a result, it is critical to hold off on ripening or entering the Rutab phase. This study determined that the date fruit ‘Zaghloul’ was subjected to varied ATP concentrations to accelerate the Rutab stage and increase the sucrose metabolism enzyme activity.

## 2. Materials and Methods

### 2.1. Date Fruits and Samples

An Egyptian date orchard in the province of Damietta was used to harvest dates for this experiment. A collection of fruits was made during the Bisir stage (fruits have become fully red). It was possible to harvest 4 bunches that were free of defects and uniform in color. A fruit sample (720 dates) was picked and divided into two main batches. The first batch (360 dates) was used for the determination of the Rutab index and physical traits. This batch was split into 4 × 90 dates each for treatment, for which there were three replicates (e.g., 3 × 30 dates). For chemical assessments and enzymes, we used the second batch with the same fruit distribution as the first. 

### 2.2. ATP: Exogenous Treatment Protocol

First, dates were treated with a sodium hydrochloride solution of 0.02 mL L^−1^ for 1 min before being cleaned with tap water. As a follow-up, dates were immersed for 10 min in the same ATP solution at doses of 0, 0.5, 1.0, and 1.5 mmol L^−1^, plus 0.01 mL L^−1^ of Tween-20 for 10 s. The ATP treatments were set with distilled water. A final step was to dry out the dates by exposing them to ambient conditions and then store them at a constant 27 ± 1 °C with relative humidity 67 ± 4 RH% for 12 d. As a control, ATP was applied at 0 mM.

### 2.3. Fruit Rutab Index

An experimenter used a magnifying glass to visually determine the Rutab Index (FR-index). Our observations during the experiment led us to categorize the FR-index into 4 categories: FR-Index 1 = no Rutab incidences; FR-Index 2 = slight changes (0–20 percent of the fruit); RU-Index 3 = moderate changes (20–50 percent of the fruit); and FR-Index 4 = full Rutab incidences (full fruit Rutab):$$\mathrm{FR}-\mathrm{index}=\overset{4}{\underset{n=4}{\sum }}\frac{\left(\mathrm{Rutab}\;\mathrm{level}\;\mathrm{No}.\right)*\left(\mathrm{No}.\;\mathrm{fruit}\;\mathrm{at}\;\mathrm{Rutab}\;\mathrm{stage}\right)}{\left(\mathrm{Total}\;\mathrm{number}\;\mathrm{of}\;\mathrm{fruits}\right)}$$

### 2.4. Soluble Solid Content (SSC%), Acidity (TA%), Moisture (%), and Tannins Content (TN)

Official methods of analysis were used to determine SSC, TA, and moisture% (AOAC, 1992). TN was measured using 100 µL of the diluted sample, 6 mL of 4% vanillin reagent in methanol solvent, and 3 mL of HCl (center) were combined to determine the total tannin content. They were blended for 15 min at room temperature and were measured at 500 nm in comparison to the solvent methanol as the blank. Three replications of each representation were performed. [23]. TN was expressed in mg (+)-catechin Kg^−1^ basis of dry weight.

### 2.5. Sucrose Metabolic: Enzyme Activities Monitoring

The fruit sample (4 g) was blended with 10 mL of 100 mM sodium phosphate (pH 7.5), magnesium chloride (10 mM), and 1 g L^−1^ polyvinylpyrrolidone (PVPP). An additional 1 mL L^−1^ Triton-X100 was also included. It was centrifuged at 4 °C for 25 min at 10,000× *g* for 20 min. They were then stored overnight at 4 °C until being measured [24].

A mixture of invertase and clear extraction enzymes (100 l each) was prepared by mixing 1000-M of sodium acetate (pH: 5.5) with 5 mM of magnesium chloride, 1 g of L^−1^ sucrose, and 1 mM of EDTA. As an afterthought, the 3,5-dinitro-salicylic acid was added to the blending solution, and it was then boiled for 5 min. Finally, the samples were kept at lab temperature to cool, and were measured sectrophotmertically at a 540 mm wavelength [25]. The AI activity was expressed as a µmol s^−1^ Kg^−1^.

NI: This method is similar to that made up in invertase (AI), but instead of sodium phosphate, it uses 1 g L^−1^ sucrose instead. The NI activity was expressed as a µmol s^−1^ Kg^−1^.

SS-s: we used 50 µL of Heps-NaOH plus 5 mM UDP and 5 mM of 0.1 of sucrose, NAF with an additional 40 µL of extraction. As with measuring AI, enzyme blend and determination of enzyme were the same. The SS-s activity was expressed as a µmol s^−1^ Kg^−1^.

SS-c: activity can be measured by mixing 4 mM UDPG with 100 mM Hepes-NaOH (pH 8.0), 15 mM MgCl_2_, 60 mM fructose, and lastly an amount of extraction enzyme. First, a 5 mM NaOH solution was added, and the mixture was heated for 5 min. The samples were also held at room temperature until cold. They were then weighed at 620 mm and incubated at 80 °C for 10 min [26]. The SS-c activity was expressed as a µmol s^−1^ Kg^−1^.

The total protein is prepared and analyzed as a particular base for calculating the catalyst activities [27].

### 2.6. Sugars Content Profile Accumulation

Inverting sugar reduces the copper in Fehling’s solution to a red, insoluble cuprous oxide, which can be measured. With Fehling’s solution as a reference, we calculated the volume of unknown sugar solution required to completely reduce a measured volume of Fehling’s solution to estimate the sugar content. To determine the glucose equivalent and the reduction factor, Fehling’s solutions A and B (5 mL of each) were standardized against standard glucose before use [1:1] [28].

Pour clarifier, we combined a 45% primary lead acetate solution with 2 g of fruit pulp and left it for 10 min at room temperature. Potassium oxalate crystals were added in excess to dilute the mixture. Finally, the mixture was diluted with distilled water and then filtered with filter paper to reach the desired volume. Using Fehling’s mixture (5 mL Fehling’s solution A + 5 mL Fehling’s solution B), the clear extract was heated until the blue faded away. This was carried out by adding 1 mL of methylene blue indicator and measuring the titer value.

This was used to determine the content of sugars in an item. For the reduction in sugar measurement, the clear mixture obtained was used. A total of 51 mL of the solution’s volume was used. For 24 h, hydrochloric acid and water were mixed and allowed to sit at room temperature. It was therefore decided to chill the contents, neutralize the solution with 40% NaOH phenolphthalein as an indicator, and increase the final volume from 50 to 100 mL. As a final step, the mixture was purified and titrated to estimate the reduction in sugar percentage.

The total sugar% was represented as a percentage in courses of inverted sugars. The non-reducing sugars were computed by multiplying the variations of total and reducing sugars by a factor of 0.95 so that the results were proved as a percentage.

### 2.7. Total Phenol (TP), Flavonoids (TFv), Polyphenol Oxidase (PPO), and Phenylalanine Ammonia-Lyase (PAL)

This enzyme activity was determined by adding one gram of fruit pulp sample in 5 mL of Tris-HCl solution, diluted to pH 7, and then mixing it. Under cooling at 4 °C, the mixture was centrifuged for 10 min at 10,000× *g*, and the clear supernatant was stored at −20 °C to record PPO activity. Catechol substrate was used to control the progress of enzyme activity. Then, 200 L of the rachis extraction was added to 3 mL of 20 mM catechol melted in 100 mM of sodium phosphate buffer, pH 7.0, in a matter of minutes [29]. Over the course of 3 min, spectrophotometric observations revealed activity at a wavelength of 400 nm. Catalysts with one unit of PPO activity make a difference of 0.10 in absorbance min. The PPO activity was expressed as a µmol s^−1^ Kg^−1^.

One gram of fruit pulp sample was added to 50 mM borate buffer (pH 8.5), containing 5 mM 2-mercaptoethanol and 400 mg PVP, to determine the phenylalanine ammonia-lyase (PAL). After centrifuging at 16,000× *g* for 15 min at 4 °C, a clear mixture was obtained. Adding 700 µL of L-phenylalanine plus 3 mL of 50 mM borate buffer to the blend produced the desired result (pH 8.5). Immediately, 300 µL of the supernatant fraction was added to it. For 60 min, the blond sat at 40 °C, stewing. It was possible to block the enzyme response by adding 100 µL of HCL (5 mM). At ambient temperature, PAL activity was estimated [30]. The PAL activity was expressed as a µmol s^−1^ Kg^−1^.

Spectrophotometric analysis of total phenolics (TP) was performed at a wavelength of 750 nm. Data were obtained and represented as mg of gallic acid equivalents (GAE) per 100 g of flavonoids [31]. Both TP and TFv were expressed as mg kg^−1^ on a fresh weight basis.

### 2.8. Malondialdehyde Concentration (MDA) and Electrolyte Leakage Percentage (EL%)

Two grams of fruit tissue were used to measure malondialdehyde (MDA). With the help of the TABRs test, the amount of lipid peroxidation in the body was determined. The homogenized mixture contained 2.5 g of berry tissue, 5% metaphosphoric acid (*w*/*v* of HPO_3_), and 2% butylhydroxytoluene (BHT) (C_15_H_24_O). As a result, a standard curve was prepared using 1,1,3,3-tetraethyoxypropane (C_7_H_16_O_4_; Sigma-Aldrich, St. Louis, MO, USA) that was comparable to 0–1 mM malondialdehyde (MDA) to estimate the MDA accumulation of date tissue during shelf-life [32]. MDA content was expressed as mmol Kg^−1^ on a fresh weight basis.

To estimate the electrical conductivity (EC) of all samples during shelf-life, samples were taken at intervals. For 3 h, 2 g of date pulp was added to 10 mL of mannitol at 6 mmol L^−1^. Next, a conductivity meter was used to measure the conductivity of the solution (M1). Following this, all cavities were boiled for an hour at 100 degrees Celsius to destroy the peel tissue. The conductivity of all cavities was then re-read as a total leakage (M2). Electrolyte leakage relative was calculated as a percentage [33].

### 2.9. Statistical Analysis

The main two-season experimental data were examined with an analysis of variance (ANOVA) test and the Tukey–Kramer HSD test to compare the different treatments. The combinations of ATP treatments and shelf-life duration in days were designed as a factorial experiment. The FR-index assay was analyzed using one-way ANOVA. There were 3 replicates per treatment, with each replicate consisting of 30 fruits. The Pearson’s correlation matrix among the parameters under study and the Principal Component Analysis (PCA) was applied. JMP Pro 16 software was used for all statistical analyses and the statistical significance was established at *p* < 0.05. To predict the properties of the treated fruits during shelf-life, linear regression was applied and R^2^ > 0.7 was considered as a good judge for the suitability of the model [34].

## 3. Results

### 3.1. FR-Index

With the progress of the shelf-life period, fruit Rutab increased continuously both in ATP-treated and control ‘Zaghloul’ fruit (Figure 1). However, the FR-Index was lower in untreated fruit than in treated fruit. After 12 d of shelf-life storage, the FR-Index of treated fruit by ATP 1.5 mmol L^−1^ was approximately fourfold higher than untreated (control). A gradual increase in fruit ripening was observed with the advancement in shelf-life periods in all treatments, but the maximum fruit Rutab process was detected in the treated fruit with 1.5 mmol ATP L^−1^ (Figure 1). Up to day 12, the FR-Index was approximately four times higher as compared with untreated fruit and other ATP treatments. On average, control fruit showed the lowest FR-index, as compared with 0.5 mmol L^−1^ ATP treatment on day 12 of shelf-life (FR-index = 1.75).

### 3.2. Change in Content SSC%, Moisture%, TA%, and TN%

Figure 2 shows that ATP dipping encouraged the increase in SSC content in Zuglaol fruit but was insignificant when compared to the untreated fruit. ATP was afforded independently to keep fruit moisture% due to the ATP concentration during shelf-life. The ATP 1.5 mmol L^−1^ treatment increased in SSC (33.25%) on the 12th day of shelf-life compared to the untreated fruit (30.32%). Furthermore, it preserved fruit maturation which was recorded (70.63%) compared to the other ATP treatment and the control fruit (55.18%) during shelf-life duration (Figure 2). However, it decreased both fruit acidity and tannin content (0.440% and 2.75 mg kg^−1^) throughout the storage period (Figure 2C,D).

### 3.3. Changes in Acid Invertase (AI), Neutral Invertase (NI), Sucrose Synthase-Cleavage (SS-c), and Sucrose Synthase-Synthesis (SS-s) Activities

For all treatments, Figure 3A shows that the AI activity of ‘Zaghloul’ fruit increased until the 3rd day, remained stable for the next 3 to 9 days, and then rose until the end of the experiment. According to Figure 2B, the fruit’s NI activity followed a similar trend and occurred at the same intervals. After the 6th day of treatment with 1.5 mmol L^−1^ of ATP, both enzyme AI and NI activity increased dramatically and reached a high level of activation on the 12th day of shelf-life time. As shown in Figure 3C,D, both SS-c and SS-s activities were affected by the addition of ATP to fruit. Due to the ATP treatments, it grew independently, reaching its peak on the 6th day, and then declining through to the end of the storage period. On the other hand, SS-c, and SS-s both increased up to the 6th day of the experiment and remained stable until the end of the storage time. On the 12th day of the shelf-life, the ATP treatment significantly increased SS-c (0.79 U) and SS-s (0.51).

### 3.4. Fruit Sugar Profile

Fruit quality is largely determined by the amount of soluble sugar and organic acid in the flesh. Sucrose and citric acid are the two major soluble sugars and organic acids associated with sweetness and acidity in flesh, respectively [35]. We observed that the soluble sugar increased with all treatments. The ATP 1.5 mmol L^−1^ increased the fruit in soluble sugar much more than the other ATP treatments and untreated fruits during the storage period (Figure 4).

### 3.5. Total Phenol (TP), Flavonoids (TFv), Polyphenol Oxidase (PPO), and Phenylalanine Ammonia-Lyase (PAL)

In dates, the brown coloration is caused by phenolic compounds, which are essential to enzyme activity. Dates ripen in four stages, known as ‘kimri’, ‘khalal’, ‘rutab’, and ‘tamer’ in Arabic (ripe, reduced moisture). As shown in Figure 5A, ATP treatments caused significant differences (*p* < 0.05) in TP, TFv, PPO, and PAL when the shelf-life duration in days and ATP treatments were considered. We observed that the ATP at 1.5 mmol L^−1^ decreased the TP and TFv during storage compared to other treatments during the shelf-life time (Figure 5).

As well as ATP treatment doses, PPO and PAL activities increased gradually and independently due to ATP treatment concentrations. Meanwhile, the ATP at 1.5 mM enhanced PPO and PA activities, which accelerated the ripening feature (Figure 5C,D).

### 3.6. Malondialdehyde Concentration (MDA) and Electrolyte Leakage Percentage (EL%)

In Figure 6, MDA concentrations (ƞmol kg^−1^) and EL% are shown for the ‘Zaghloul’ date. Based on time in storage, the differences were charted in days. After considering the effects of time and ATP doses, the MDA and EL% showed a significant interaction with each other. It was found that the rate of MDA products increased as ATP concentrations increased from 0.5 to 1.5 mmol L^−1^ in fruits immersed in the solution (Figure 5A). A similar reaction was observed when compared to other ATP treatments and control fruits, with the EL% being the greatest (Figure 6B).

### 3.7. Multivariate Analysis of Date Parameters

A PCA for the physiological and biochemical parameters of the data collected from the date stored under shelf-life conditions was carried out for the affected ATP at different concentrations. Fruits were stored at 27 ± 1 °C with relative humidity 67 ± 4 RH% for 12 d. The PCA separated the effect of ATP on the date fruits under shelf-life conditions. The PC1 explained 82.3% of the variability in the data, while PC2 explained 7.57% variability (Figure 7A). Figure 7B shows the negative correlation between the FR-index and all the variables. The sucrose enzyme activities were positively correlated with the FR-index. These four variables (TP, TFv, PPO, and PAL) had a positive correlation with the FR-index and showed a negative correlation with MDA and EL%. The FR-index was positively correlated with AI, NI, SS-s, and SS-c, whereas it had a negative correlation with other values such as TN. Pearson’s correlation matrix, among the studied parameters, shows the correlation and indicates these results (Table 1).

### 3.8. Modelling and Validation

Linear regression analysis was performed to predict the properties of the treated fruit with the best treatment (ATP 1.5 mM). Obviously, it can be seen from Table 2 that the R^2^ values of the model for the listed properties are greater than 0.70, except for SS-c, meaning that the linear model could be satisfactorily used for predicting the properties of the treated date with ATP 1.5 mM at different storage periods from 0 to 12 days. The linear model cannot be beneficial for forecasting SS-c as its R^2^ is less than 0.7.

## 4. Discussion

To keep fruits and vegetables fresh for as long as possible after harvest, they must be kept fresh in terms of their aroma and texture as well as their flavor and ascorbic acid. Dehydration affects taste, nutrition, and flavor and can lead to the decline of sensory quality [36]. The Rutab stage of ‘Zaghloul’ begins to ripen at the apex, turning brown or black in color and becoming soft. Polygalacturonase, beta-galactosidase, and cellulose enzymes all play a role in date fruit softening [6,37]. Softening causes the tannins under the skin to precipitate in an insoluble form, resulting in the fruit losing astringency and gaining total sugars and total solids concentrations [38]. This experiment exhibited that ATP application significantly increased the fruit FR-index of ‘Zaghloul’ fruit (Figure 1). The ATP synthesis in cells relies on respiration, which is controlled by the cell’s energy status. Exogenous ATP has been shown to reduce respiration by improving energy status in previous studies [39,40]. In addition to this, our results showed that the level of energy charge had a negative effect on the synthesis of ATP [22]. The senescence of the fruit is inextricably linked to energy deficiency [19].

Date fruits contain a large content of tannins, which cause the astringent taste of the fruits in the Biser stage. These substances decrease with the increasing maturity of the fruits and reduce completely at the Rutab stage. At the beginning of storage, the organic acid is converted into nutrients such as soluble sugars, amino acids, and vitamins by the application of ATP to the fruit. We observed that the ATP treatment of 1.5 mmol L^−1^ increased SSC% compared with the control during the entire shelf-life period (Figure 2A), which agrees with the results approved previously [19,39]. Changes in energy levels may be a key factor in fruit senescence, according to all of these findings [36]. Litchis [18], longans [41], Nanguo pears [19], and mung bean sprouts [21] have been shown to benefit from exogenous ATP treatment.

A crucial biological process, sucrose syntheses can enhance the tissue defenses against biotic and abiotic stresses [42,43]. By accumulating sugar and maintaining a high energy level in fruits, softening can be delayed [44]. Research has shown the role of sugar and energy in maintaining the quality in loquats and Chinese bayberry fruit, as well as improving chilling tolerance in apricots [45] and peaches [46]. Furthermore, SS-c is catalyzed by AI, NI, and SS-c in sucrose metabolism, whereas SS-s is responsible for sucrose synthesis from glucose and fructose [47].

Not only does sugar play a crucial role in fruit growth and development, but it also serves as a substrate for various metabolic processes in fruits, as well as a source of energy. Sugar accumulation and fruit quality are closely related to enzyme activity in sucrose metabolism [48,49]. Moreover, from the above results it could be concluded that ATP treatment could effectively inhibit the decrease in SSC and total soluble sugars [20], and retard to maintain quality at the Rutab stage.

Due to the ripening processes generated, the phenolic compound decreases and the browning enzyme activity increases after harvesting [50]. It is possible that a dose of ATP at a rate of 1.5 mmol L^−1^ can accelerate the ripening process of the fruits at room conditions, which would break down the pectin during ripening [21]. Furthermore, this encourages the activity of the brown pigmentation enzymes associated with the ripening process [51]. In addition, it could be that the ATP after the 6th day of shelf-life enhanced ripening processes much more, presenting Rutab trails on the 12th day of storage. Our results are in agreement with the previous study on logan fruit [52].

In this paper, we explained the differences in MDA and EL% following ATP treatments. As a result of increased intracellular potential charge after injecting ATP [19], MDA and EL% were recorded throughout storage [39]. Therefore, cell membrane lipid/fatty acid compartments were preserved, which resulted in a lower EL% [53,54]. Due to this, the use of ATP helped to maintain a high standard of quality [19]. Fruit quality was also maintained by delaying fruit senescence and reducing ethylene production during storage [18]. We also found out that by reducing fruit tissue breakdown, ATP improved the quality of the fruit as well. Our results, on the other hand, contradict other findings in the period from the sixth to the twelfth day of shelf-life duration.

In conclusions, the optimal concentration of ATP immersions could improve sucrose enzyme activities and decrease certain bioactive compounds, such as total phenolic compounds and tannin, of the fruit during storage. Interestingly, ATP treatments, especially at 1.5 mM, enhanced the concentrations of total sugar during storage. This study suggests that high-dose ATP immersion, especially at 1.5 mmol L^−1^, is a feasible, potential treatment to encourage fruit to reach the Rutab stage during storage. The linear model could be satisfactorily used for predicting the properties of the treated date with ATP 1.5 mM at different storage periods from 0 to 12 days. Furthermore, it is a new potential tool that pushes the fruits to regular ripening after harvest, thus reducing the loss in fruits during the production process.

## Figures and Tables

**Figure 1 foods-10-02641-f001:**
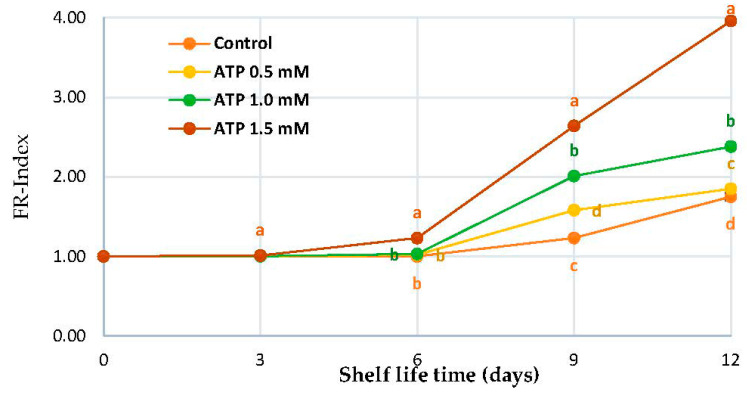
FR-index of ‘Zaghloul’ date fruit treated with ATP at different concentrations (0, 0.5, 1.0, and 1.5 mmol L^−1^) during storage at 27 ± 1 °C, 67 ± 3% R, for 12 d. Each value represents mean ± SE (*n* = 3). The letters differ (*p* < 0.05) and represent the significance between treatments using the Tukey–Kramer HSD test to compare the different treatments.

**Figure 2 foods-10-02641-f002:**
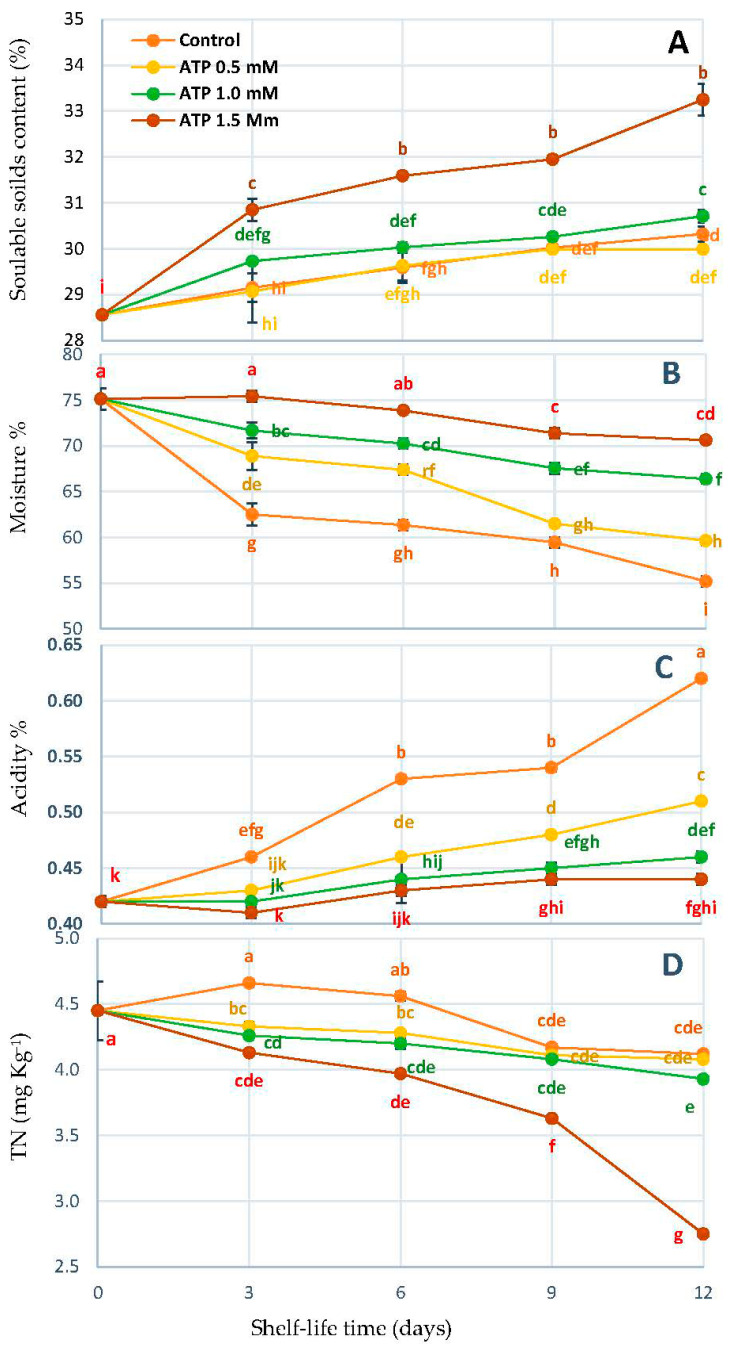
SSC% (**A**), moisture% (**B**), TA% (**C**), and TN (**D**) of ‘Zaghloul’ date fruit treated with ATP at different concentrations (0, 0.5, 1.0, and 1.5 mmol L^−1^) during storage at 27 ± 1 °C, 67 ± 3% R, for 12 d. Each value represents mean ± SE (*n* = 3). The letters differ (*p* < 0.05) and represent the significance between treatments using the Tukey–Kramer HSD test to compare the different treatments.

**Figure 3 foods-10-02641-f003:**
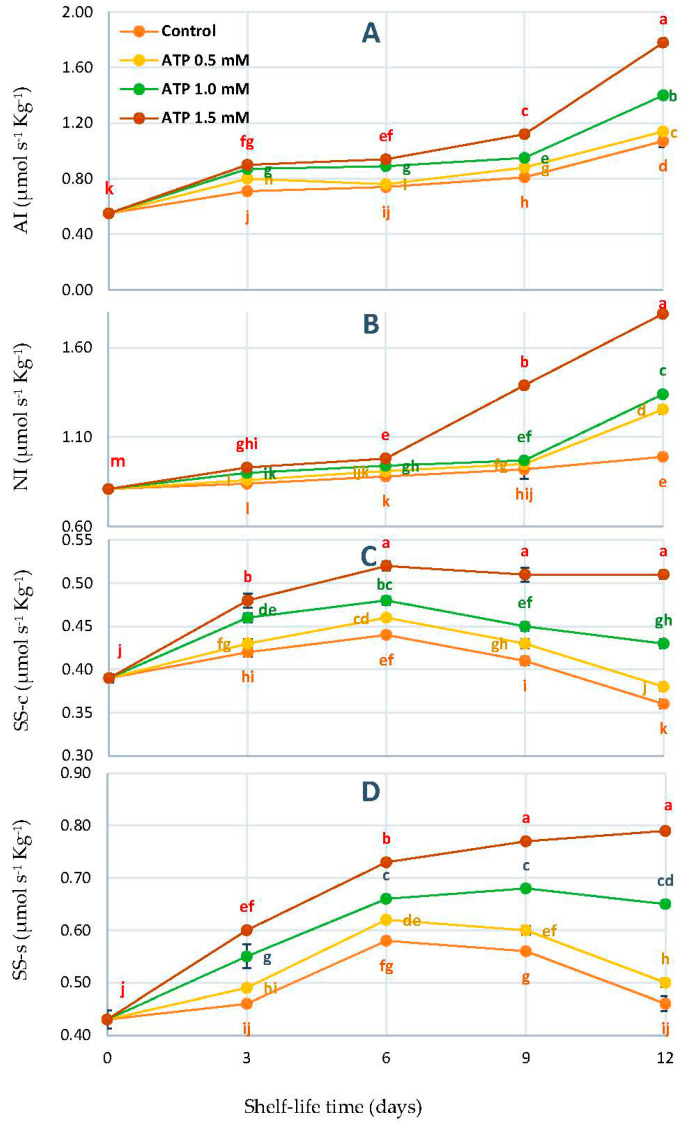
AI (**A**), NI (**B**), SS-c (**C**), and SS-s (**D**) of ‘Zaghloul’ date fruit treated with ATP at different concentrations (0, 0.5, 1.0, and 1.5 mmol L^−1^) during storage at 27 ± 1 °C, 67 ± 3% R, for 12 d. Each value represents mean ± SE (*n* = 3). The letters differ (*p* < 0.05) and represent the significance between treatments using the Tukey–Kramer HSD test to compare the different treatments.

**Figure 4 foods-10-02641-f004:**
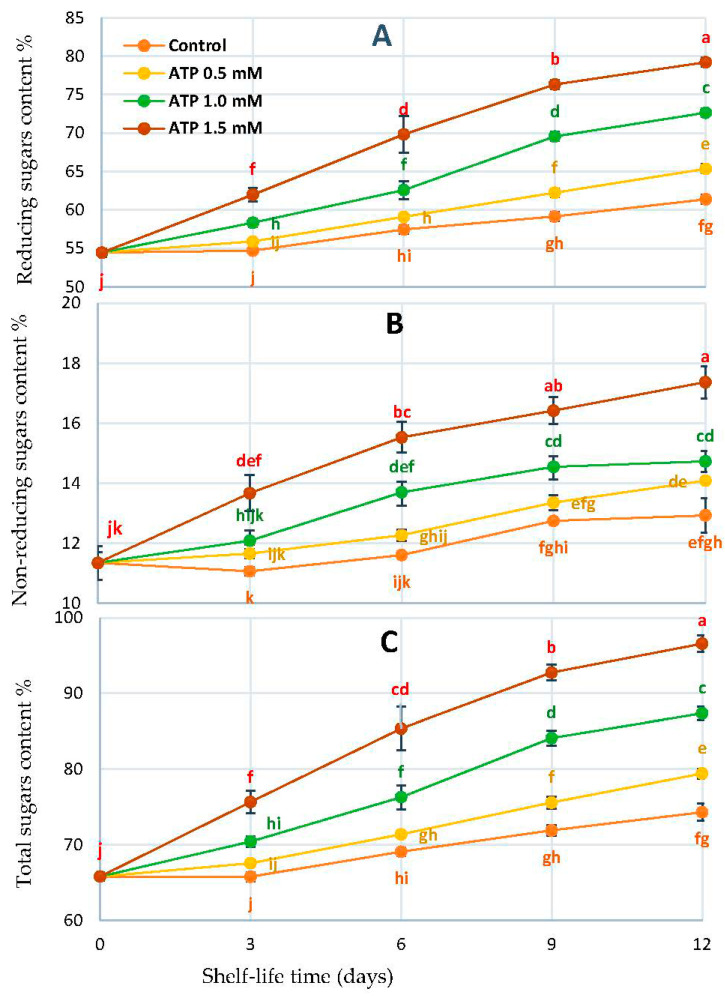
Reducing sugar (**A**), non-reducing sugar (**B**), and total soluble sugar (**C**), of ‘Zaghloul’ date fruit treated with ATP at different concentrations (0, 0.5, 1.0, and 1.5 mmol L^−1^) during storage at 27 ± 1 °C, 67 ± 3% R, for 12 d. Each value represents mean ± SE (*n* = 3). The letters differ (*p* < 0.05) and represent the significance between treatments using the Tukey–Kramer HSD test to compare the different treatments.

**Figure 5 foods-10-02641-f005:**
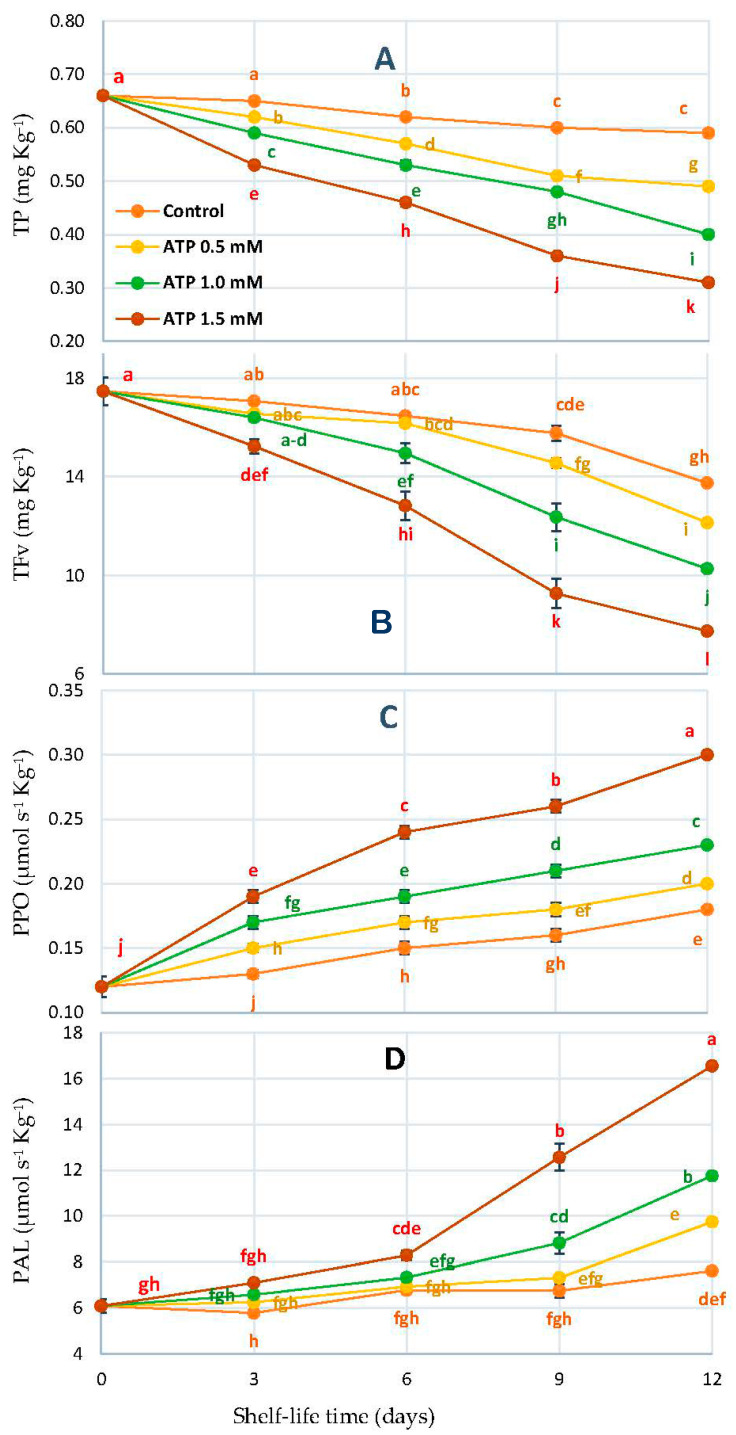
TP (**A**), TFv (**B**), PPO (**C**), and PAL (**D**) of ‘Zaghloul’ date fruit treated with ATP at different concentrations (0, 0.5, 1.0, and 1.5 mmol L^−1^) during storage at 27 ± 1 °C, 67 ± 3% R, for 12 d. Each value represents mean ± SE (*n* = 3). The letters differ (*p* < 0.05) and represent the significance between treatments using the Tukey–Kramer HSD test to compare the different treatments.

**Figure 6 foods-10-02641-f006:**
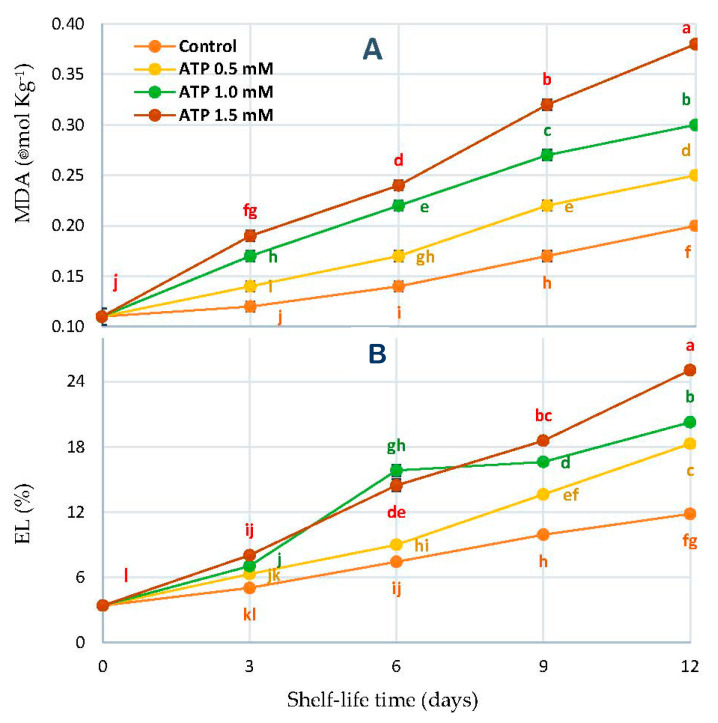
MDA (**A**) and EL% (**B**) of ‘Zaghloul’ date fruit treated with ATP at different concentrations (0, 0.5, 1.0, and 1.5 mmol L^−1^) during storage at 27 ± 1 °C, 67 ± 3% R, for 12 d. Each value represents mean ± SE (*n* = 3). The letters differ (*p* < 0.05) and represent the significance between treatments using the Tukey–Kramer HSD test to compare the different treatments.

**Figure 7 foods-10-02641-f007:**
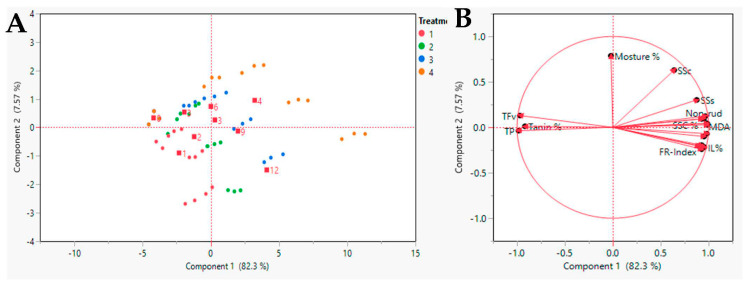
Presents the Principal Component Analysis (PCA) representing shelf-life duration in days and ATP (four) to ‘Zaghloul’ date, plotted with the contribution of each parameter on the four PCA axes (**A**), and all the physiological and biochemical parameters measured in fruit during the storage period (**B**). Principal Component Analysis (PCA) variable correlation.

**Table 1 foods-10-02641-t001:** Pearson’s correlation matrix among the studied parameters of ‘Zaghloul’ date response to ATP treatments during shelf-life.

	FR-Index	SSC%	Moisture%	TA%	TN	AI	NI	SS-s	SS-c	RS	NRS	Total S	TP	TFv	PPO	PAL	MDA	EL%
**FR-Index**	* 1.0000																	
**SSC%**	0.7652	1.0000																
**Moisture%**	−0.0422	−0.0338	1.0000															
**TA%**	0.8632	0.9046	0.1080	1.0000														
**TN**	−0.8413	−0.8795	0.0230	−0.9038	1.0000													
**AI**	0.8673	0.8480	−0.1505	0.8648	−0.8920	1.0000												
**NI**	0.9423	0.8176	−0.0538	0.8906	−0.8666	0.9396	1.0000											
**SS-s**	0.6699	0.8524	0.0308	0.8515	−0.7842	0.6962	0.6798	1.0000										
**SS-c**	0.3772	0.6987	0.2425	0.6745	−0.6011	0.4327	0.4350	0.8572	1.0000									
**RS**	0.8557	0.8906	0.0016	0.9443	−0.8712	0.8781	0.8796	0.8746	0.6277	1.0000								
**NRS**	0.7900	0.8887	0.0450	0.9199	−0.8069	0.8248	0.8202	0.8637	0.6365	0.9471	1.0000							
**Total S**	0.8498	0.8978	0.0104	0.9474	−0.8657	0.8749	0.8752	0.8799	0.6349	0.9979	0.9658	1.0000						
**TP**	−0.8574	−0.8942	−0.0061	−0.9295	0.8679	−0.8907	−0.8990	−0.8826	−0.6529	−0.9795	−0.9441	−0.9808	1.0000					
**TFv**	−0.8798	−0.8618	0.0839	−0.9084	0.8312	−0.9115	−0.9148	−0.7891	−0.4959	−0.9660	−0.9500	−0.9710	0.9577	1.0000				
**PPO**	0.8124	0.9323	−0.0302	0.9399	−0.8801	0.9014	0.8705	0.8871	0.6704	0.9663	0.9659	0.9745	−0.9688	−0.9586	1.0000			
**PAL**	0.9424	0.8347	0.0128	0.9342	−0.8816	0.9173	0.9746	0.7431	0.4997	0.9209	0.8664	0.9178	−0.9172	−0.9390	0.9000	1.0000		
**MDA**	0.8703	0.8791	−0.0647	0.9128	−0.8741	0.9236	0.8996	0.8515	0.5629	0.9721	0.9446	0.9749	−0.9801	−0.9743	0.9702	0.9228	1.0000	
**EL%**	0.8714	0.8383	−0.1864	0.8778	−0.8670	0.9284	0.9029	0.7837	0.4617	0.9479	0.8929	0.9449	−0.9508	−0.9590	0.9298	0.9088	0.9737	1.0000

* Values express average values per shelf-life duration in days (12 d), and four ATP treatments. FR-Index—fruit rutab index; SSC%—total soluble solids; Moisture%—fruit moisture content; TA%—acidity; TN—soluble tannin content; AI—acid invertase; NI—neutral invertase; SS-c—SS-cleavage; SS-s—sucrose synthase; RS—reducing sugar; NRS—non-reducing sugar; TS—total sugar; TP—total phenol; TFv—flavonoids content; PPO—polyphenol oxidase; PAL—phenylalanine ammonia-lyase; MDA—malondialdehyde accumulation; EL%—electrolyte leakage percentage. The different color values point out the positive or negative correlation among variables.

**Table 2 foods-10-02641-t002:** Modelling of changes in the properties during shelf-life of the treated dates with ATP 1.5 mM.

Properties	Linear Model (Y = a ± bX) *		
	a (*p*-Value)	b (*p*-Value)	R^2^
**FR-Index**	0.459 (0.00)	0.252 (0.064)	0.836
**SSC%**	29.15 (0.00)	0.349 (0.00)	0.886
**Moisture%**	75.92 (0.00)	−0.436 (0.00)	0.729
**TA%**	0.418 (0.00)	0.016 (0.00)	0.95
**TN**	4.75 (0.00)	−0.151 (0.00)	0.892
**AI**	0.526 (0.00)	0.089 (0.00)	0.874
**NI**	0.695 (0.00)	0.081 (0.00)	0.892
**SS-s**	0.49 (0.00)	0.029 (0.00)	0.872
**SS-c**	0.43 (0.00)	0.009 (0.00)	0.588
**RS**	55.6 (0.00)	2.13 (0.00)	0.94
**NRS**	11.9 (0.00)	0.493(0.00)	0.848
**Total S**	67.49 (0.00)	2.624 (0.00)	0.936
**TP**	0.639 (0.00)	0.029 (0.00)	0.971
**TFv**	17.59 (0.00)	0.848 (0.00)	0.948
**PPO**	0.138 (0.00)	0.0141 (0.00)	0.942
**PAL**	4.89 (0.00)	0.886 (0.00)	0.906
**MDA**	0.116 (0.00)	0.0218 (0.00)	0.978
**EL%**	3.14 (0.00)	0.79 (0.00)	0.984

* Y and X denote the dependent (properties) and independent (shelf-life duration) variables, respectively.

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
