# Peer review of "Enhance Fruit Ripening Uniformity and Accelerate the Rutab Stage by Using ATP in ‘Zaghloul’ Dates during the Shelf Life"

_foods, 2021, doi:10.3390/foods10112641_

Round 1

Reviewer 1 Report

Line 18: correct „Fculty“ for „Faculty“.

ABSTRACT section

Lines 25-27: „The objective of...“ is an unclear sentence; it should be reformulated.

Line 27: at the beginning of the sentence it is said that the untreated fruits were dipped only in water. Therefore, it is not necessary to state, as given in the continuation of the sentence, that the concentration of ATP is 0, but only to state that the concentrations of ATP are 1.0 and 1.5.

Line 29: correct „ATP immersion“ for „immersion in the ATP solution“.

Lines 29-31: an unclear sentence; it should be reformulated. Especially part „could induce (induce what?)...certain biologically active compounds...“.

Line 33: correct „perentage“ for „percentage“.

Line 34: what is the meaning of the abbreviation AI (acid invertase? or artificial invertase?). This should be stated before the brackets.

Line 35: „cleavage SS“ -  this needs to be better explained (before the brackets) – i.e. what is „cleavage SS“.

Lines 35-36: „Interestingly, 1.5 mM ATP immersion forced the date fruit to reach the Rutab stage during storage“, should be corrected as follows: „Interestingly, immersion in 1.5 mM ATP  forced the date fruit to reach the Rutab stage during storage“.

Lines 36-40: Sentences are not clear. They have to be reformulated.

INTRODUCTION section

Line 48-64: it is necessary to better explain the stages of date fruit maturity. See  http://dx.doi.org/10.1007/978-3-030-06120-3_6-1

Lines 64-66: Is it true that „loosing water....resulting in softening“ (??)

Lines 66-69: Sentences are not clear. They have to be reformulated.

In general, the Introduction should be corrected to make it easier to see the importance of research, to better explain the individual stages of date ripeness, and the importance of applying ATP solution during the ripening process.

MATERIALS AND METHODS section

Line 94: According http://dx.doi.org/10.1007/978-3-030-06120-3_6-1 the fruits in Khalal stage is yellowish!!!!!.

Line 115: SSC, TA, TN are mentioned for the first time in text. It should be explained what these abbreviations mean.

In general, the Materials and methods section should be revised in terms of language, better explanations of some abbreviations, and more precise writing of text (for example Line 176 – is it true that „...200 l (??????) of the rachis extraction....“ that was used; There are a lot of imprecise and insufficiently clear parts of the text).

RESULTS section

Lines 222-223: correct „4 fold higher“ for „about 4 fold higher“.

Lines 226-227: similar comment as comment above.

In general, the Results section contains very interesting results of researching. However, the textual part must be written better, in terms of language and precision. Also, legends in figures as well as designations of units on the y-axes of figures should be accurately written.

DISCUSSION section

Discussion in some aspects must be improved to be publishable.

Author Response

Manuscript ID

foods-1420482

Type Article

Title: Enhance fruit ripening uniformity and accelerate the Rutab stage by using ATP in ‘Zaghloul’ Dates During the Shelf Life

Author's Reply to the Review Report (Reviewer 1)

Line 18: correct „Fculty“ for „Faculty“.

Corrected

ABSTRACT section

 Lines 25-27: „The objective of...“ is an unclear sentence; it should be reformulated.

 Rephrased to be clear

Line 27: at the beginning of the sentence it is said that the untreated fruits were dipped only in water. Therefore, it is not necessary to state, as given in the continuation of the sentence, that the concentration of ATP is 0, but only to state that the concentrations of ATP are 1.0 and 1.5.

Agree with you, it changed

Line 29: correct „ATP immersion“for „immersion in the ATP solution“.

It was added

Lines 29-31: an unclear sentence; it should be reformulated. Especially part „could induce (induce what?)...certain biologically active compounds...“.

Rephrased

Line 33: correct „perentage“ for „percentage“.

Corrected

Line 34: what is the meaning of the abbreviation AI (acid invertase? or artificial invertase?). This should be stated before the brackets.

acid invertase (AI), neutral invertase (NI), sucrose synthase-cleavage (SS-c), and sucrose synthase-synthesis  (SS-s) 

Line 35: „cleavage SS“ -  this needs to be better explained (before the brackets) – i.e. what is „cleavage SS“.

 acid invertase (AI), neutral invertase (NI), sucrose synthase-cleavage (SS-c), and sucrose synthase-synthesis  (SS-s)

Lines 35-36: „Interestingly, 1.5 mM ATP immersion forced the date fruit to reach the Rutab stage during storage“, should be corrected as follows: „Interestingly, immersion in 1.5 mM ATP  forced the date fruit to reach the Rutab stage during storage“.

Changed and added you comment

Lines 36-40: Sentences are not clear. They have to be reformulated.

INTRODUCTION section

Line 48-64: it is necessary to better explain the stages of date fruit maturity. See http://dx.doi.org/10.1007/978-3-030-06120-3_6-1

Explained and added

Lines 64-66: Is it true that „loosing water....resulting in softening“ (??)

Agree it changed to fruit weight loss

Lines 66-69: Sentences are not clear. They have to be reformulated.

 Rephrased

In general, the Introduction should be corrected to make it easier to see the importance of research, to better explain the individual stages of date ripeness, and the importance of applying ATP solution during the ripening process.

We add a  main problem at the

MATERIALS AND METHODS section

Line 94: According http://dx.doi.org/10.1007/978-3-030-06120-3_6-1 the fruits in Khalal stage is yellowish!!!!!.

Thank you for your comment; our cultivar is complete red to before Rutab stage … Corrected

Line 115: SSC, TA, TN are mentioned for the first time in text. It should be explained what these abbreviations mean.

Corrected and explained

In general, the Materials and methods section should be revised in terms of language, better explanations of some abbreviations, and more precise writing of text (for example Line 176 – is it true that „...200 l (??????) of the rachis extraction....“ that was used; There are a lot of imprecise and insufficiently clear parts of the text).

Corrected and explained

RESULTS section

Lines 222-223: correct „4 fold higher“ for „about 4 fold higher“.

Corrected

Lines 226-227: similar comment as comment above.

Corrected

 In general, the Results section contains very interesting results of researching. However, the textual part must be written better, in terms of language and precision. Also, legends in figures as well as designations of units on the y-axes of figures should be accurately written.

All figure and abbreviations were checked

DISCUSSION section

Discussion in some aspects must be improved to be publishable.

This part was checked

Thank you for your efforts, and my best regards

Reviewer 2 Report

The presented study is focused to enhance fruit ripening uniformity and accelerating the Rutab stage using ATP. Although the manuscript is generally well written and contains some new knowledge, I have some comments and remarks.
Authors should explain all abbreviations before first time using.
The statistic analysis should be a point 2.8 =, not “a”.
In Table 1 authors included the correlation coefficients with different colors. These different colors should be explained below the table.
Tables 2  includes the linear equations with determined coefficients of determination. Please include also a significance level for these correlations.
In conclusion, the authors stated that ATP treatments, especially at 1.5 mM, enhanced the concentrations of total sugar during storage. This study suggests that high-dose ATP immersion, especially at 1.5 mmolL-1, is a feasible potential treatment to enforce fruit to 671 the Rutab stage during storage. Why authors did nit not use the higher or optimum concentration ATP. Maybe it would be better?
Line 625: Why in this line is the “5. Conclusion” is included?

Author Response

Author's Reply to the Review Report (Reviewer 2)

The presented study is focused to enhance fruit ripening uniformity and accelerating the Rutab stage using ATP. Although the manuscript is generally well written and contains some new knowledge, I have some comments and remarks.

Authors should explain all abbreviations before first time using.

Explained

The statistical analysis should be a point 2.8 =, not “a”.

 Formatted and inserted

In Table 1 authors included the correlation coefficients with different colors. These different colors should be explained below the table.

Agree, added for distinguished among the variables positively or negatively correlation

Tables 2 includes the linear equations with determined coefficients of determination. Please include also a significance level for these correlations.

Added

In conclusion, the authors stated that ATP treatments, especially at 1.5 mM, enhanced the concentrations of total sugar during storage. This study suggests that high-dose ATP immersion, especially at 1.5 mmolL-1, is a feasible potential treatment to enforce fruit to 671 the Rutab stage during storage.

Agree

Why authors did nit not use the higher or optimum concentration ATP.

Maybe it would be better? Applied

Line 625: Why in this line is the “5. Conclusion” is included? Immerge with discussion as a new paragraph

Thank you for your effort 

Reviewer 3 Report

Dear Authors,

the topic of this article is interesting but I have a general objection to the style. You use too many abbreviations without a previous explanation (full name). Also, materials and methods are a little bit confused, too many explanations, concentration, units, measurements... It is difficult to understand whole procedures (experiment design is also uncleared, how many groups of dates). Inside the article, you can see comments and suggestions

Best regards

Author Response

Author's Reply to the Review Report (Reviewer 3)

Dear Authors,

The topic of this article is interesting but I have a general objection to the style. You use too many abbreviations without a previous explanation (full name). Also, materials and methods are a little bit confused, too many explanations, concentration, units, measurements... It is difficult to understand whole procedures (experiment design is also unclear, how many groups of dates). Inside the article, you can see comments and suggestions

Best regards

All comments in the PDF attached were answered 

  • Gibberellin not GA3 in line 67
  • It is a little bit confused. Requires better explanation how many groups in Leans 97-99

       Formatted to be clear

  1. It is a little bit confused. Requires better explanation how many groups

      Explained 2.4.line 115

  1. Line 138 mm to mM
  2. Line 139 min to minute
  3. Line 141 AI to invertase
  4. Line 145 540 mm measurements  The sentence was formatted
  5. Line  147 Sentence - instead twice mentioned Formatted
  6. Line  156 What means lab and   cold? Which temperature? Sorry, it is mistake, it changed
  7. This part is confused and has too much data. If there is a standard or official method it is enough to mention it as reference. If you have modified method it needs better explanation This parts Rephrased
  8. Line 543 mandarin to date
  9. Line 544 PCA to Principal Component Analysis
  10. Line 680 Delete the 5.Conclusion deleted
  11. Line 643 dose sugar Corrected
  12. Line 934 TSS, and total soluble sugars Miss typing error changed to SSC
  13. Line 954 delete the word compound

Round 2

Reviewer 2 Report

The authors improved the manuscript as suggested.